# The Aryl Hydrocarbon Receptor: Impact on the Tumor Immune Microenvironment and Modulation as a Potential Therapy

**DOI:** 10.3390/cancers16030472

**Published:** 2024-01-23

**Authors:** Brian D. Griffith, Timothy L. Frankel

**Affiliations:** 1Department of Surgery, University of Michigan, Ann Arbor, MI 48109, USA; briangr@med.umich.edu; 2Rogel Cancer Center, University of Michigan, Ann Arbor, MI 48109, USA

**Keywords:** aryl hydrocarbon receptor, tumor immune microenvironment, immunotherapy, tumor immune evasion

## Abstract

**Simple Summary:**

The aryl hydrocarbon receptor (AhR) is a cytoplasmic and environmental receptor that responds to both exogenous and endogenous ligands to impart a broad range of functions and thereby significantly impact cancer progression. AhR activation impacts both tumor-intrinsic pathways and immune cells in a context-specific manner. Understanding the impact of AhR activation on the tumor immune microenvironment is critical to guide cancer therapies targeting the receptor.

**Abstract:**

The aryl hydrocarbon receptor (AhR) is a ubiquitous nuclear receptor with a broad range of functions, both in tumor cells and immune cells within the tumor microenvironment (TME). Activation of AhR has been shown to have a carcinogenic effect in a variety of organs, through induction of cellular proliferation and migration, promotion of epithelial-to-mesenchymal transition, and inhibition of apoptosis, among other functions. However, the impact on immune cell function is more complicated, with both pro- and anti-tumorigenic roles identified. Although targeting AhR in cancer has shown significant promise in pre-clinical studies, there has been limited efficacy in phase III clinical trials to date. With the contrasting roles of AhR activation on immune cell polarization, understanding the impact of AhR activation on the tumor immune microenvironment is necessary to guide therapies targeting the AhR. This review article summarizes the state of knowledge of AhR activation on the TME, limitations of current findings, and the potential for modulation of the AhR as a cancer therapy.

## 1. Introduction

The aryl hydrocarbon receptor (AhR) is a cytoplasmic receptor and ligand-dependent transcription factor that functions as an environmental sensor. Through binding with a broad spectrum of both endogenous and exogenous ligands, AhR activates or inhibits cellular pathways in a cell-specific and context-specific manner [1]. Table 1 details common agonists and antagonists of AhR and their sources, which have previously been reviewed in detail [2,3,4]. In the absence of ligands, AhR is located in the cytoplasm, and activation through binding with a ligand leads to nuclear translocation and interaction with the AhR nuclear translocator (ARNT) to bind dioxin or aryl hydrocarbon response elements and promote transcriptional regulation (Figure 1) [1,5], as detailed in depth in the review by Larigot et al. [6]. AhR was initially studied as a receptor for the exogenous ligand 2,3,7,8-tetrachlorodibenzo-*p*-dioxin (TCDD) [7] and shown to have carcinogenic effects in a variety of organs [8]. Through direct activation of cancer cells, AhR signaling has been implicated in induction of cellular proliferation and migration [9,10], promotion of epithelial-to-mesenchymal transition and metastasis [11], and the inhibition of apoptosis [12].

In addition to revealing their broad role in carcinogenesis, initial studies on AhR and TCDD demonstrated an immunosuppressive effect of TCDD on T cells and dendritic cells [13]. More recently, AhR activation has been shown to have a physiologic role in the differentiation and maintenance of the function of immune cells upon binding to ligands from diet, microbial byproducts, and host cell metabolism [14]. With significant effects on immune function and its direct signaling in tumor cells, activation of AhR has emerged as an important modulator of the tumor immune microenvironment, with both pro- and anti-tumor effects.

Multiple strategies have been proposed to modulate AhR activity, including regulation of endogenous ligand production. Inhibition of indoleamine-2,3-dioxygenase 1 (IDO1), a key enzyme regulating tryptophan catabolism to the AhR ligand (AhRL) kynurenine, has been proposed to abrogate tumor intrinsic malignant signaling and enhance anti-tumor immunity [15,16]. Preclinical studies of IDO1 in experimental models of cancer using this strategy have shown promise [17,18,19]. However, despite the theoretical benefit of AhR modulation, phase III clinical trials to date have been ineffective—inhibition of IDO1 in combination with immune checkpoint inhibitors (ICI) showed no difference in survival compared to ICI alone in metastatic or unresectable melanoma [20]. Recently, the discovery of the enzyme interleukin-4-induced-1 (IL4I1) as an additional regulator of tryptophan catabolism with IDO1 [21], along with the likely paracrine pathways of AhR production within tumors [22], suggested that inhibition of IDO1 alone may be insufficient to target AhR signaling. Additionally, given the complex cell-specific role of AhR signaling on immune function within the tumor microenvironment (TME), therapeutic targeting of AhR in cancer requires careful consideration. Herein, we review the impact of AhR on the tumor immune microenvironment, limitations of current findings, and the potential for modulation of AhR as a cancer therapy.

## 2. Role in Carcinogenesis

AhR has long been associated with a variety of cancers, and its expression is elevated and the receptor chronically active in tumors, including T cell leukemia [23], B cell lymphoma [24], hepatocellular carcinoma [25], glioblastoma [26], and lung cancer [27], among others. Wang et al. previously reviewed the association of AhR activity, increased tumor aggression, and worse oncologic outcomes [28]. Constitutive activation is thought to impact carcinogenesis through disruptions in physiologic and pathologic processes, including cellular proliferation and migration, apoptosis, extracellular matrix remodeling, and the induction of angiogenesis (Figure 1).

Elevated AhR activity leads to a variety of downstream tumor-intrinsic effects that impact tumor proliferation and therapeutic resistance, including modulation of cellular proliferation and migration, epithelial-to-mesenchymal transition, apoptosis, angiogenesis, stemness, and expression of immune checkpoints.

### 2.1. Cellular Proliferation and Migration

AhR activation has been implicated in cellular proliferation in a variety of tumor types. Disruption of AhR function in a murine hepatoma cell line led to decreased proliferation and prolonged doubling time compared to its wild-type counterpart [29]. Overactivation by treatment with the AhR agonist TCDD led to increased cellular proliferation in a lung cancer cell line [30], and a similar trend was seen in human mammary epithelial cells [10].

In addition to playing a role in cellular turnover, AhR signaling appears to influence the propensity of tumors to invade and migrate through modulation of cell adhesion, where the loss of cell adhesion molecules and signaling renders cancer cells more motile and invasive [31]. AhR activation with TCDD disrupted cell–cell contact and induced migration via a c-Jun N-terminal kinase (JNK)-dependent pathway in breast cancer [9], which was abrogated with JNK inhibition [32]. This AhR-dependent cellular migration appears to be mediated in part by regulation of matrix metalloproteinases (MMPs) and mediator of cell motility 1 (Memo-1). In gastric cancer, AhR activation via TCDD induces MMP-9 expression and enzymatic activity in a dose-dependent manner, thereby promoting cellular invasion [33]. Studies in a prostate cancer model corroborated the upregulation of MMP-9 expression with AhR activity [34]. In colon cancer, Ahr/ARNT activity was mediated by expression of Memo-1, a gene implicated in colorectal carcinoma (CRC) migration, and Memo-1 depletion led to decreased CRC cell migration and invasion [35]. These studies suggest that AhR activation influences loss of cell–cell contact and extracellular matrix remodeling, thereby promoting cellular migration and invasion.

### 2.2. Epithelial-to-Mesenchymal Transition and Metastasis

Epithelial-to-mesenchymal transition (EMT) is the process through which epithelial cells, which normally interact with a basement membrane, lose apicobasal polarity and acquire a more mesenchymal phenotype, thereby increasing the ability of cells to metastasize [11]. Loss of E-cadherin expression is an important step in EMT, which is regulated by the genes Snail, Slug, Twist, and vimentin [11,36]. In both triple-negative breast cancer and inflammatory breast cancer cell lines, AhR activation via 6-formylindolo(3,2-b)carbazole (FICZ), a prototypical AhR ligand, increased expression of the EMT-associated genes Snail1, Twist, and vimentin [37]. Similarly, induction of an AhR plasmid into mammary epithelial cells induced motility, vimentin expression, and morphologic changes consistent with EMT [10]. Clinically, high AhR expression in inflammatory breast cancer correlates with lymph node metastases and advanced tumor grade [38]. This induction of EMT has also been found in other tumor types. AhR was overexpressed in esophageal squamous cell carcinoma (SCC), and selective AhR modulation with DIM (3,3′-diindolylmethane) inhibited migration and invasion and downregulated the mesenchymal markers vimentin and Slug to suppress both EMT and metastases [39]. Taken together, these studies implicate a role for AhR activation in the promotion of EMT and the propensity for tumors to metastasize.

### 2.3. Apoptosis

One of the hallmarks of malignant cells is the ability to resist cell death and apoptosis. Recent studies have implicated the AhR pathway in resistance to cellular death through upregulation of anti-apoptotic protein expression. Treatment with TCDD led to the loss of apoptosis response in three lymphoma cell lines through the downregulation of B-cell lymphoma-extra-large (Bcl-xL) and myeloid cell leukemia-1 (mcl-1), which was abrogated with an AhR antagonist [12]. In skin keratinocytes, AhR activation was shown to decrease UVB-induced apoptosis [40], and AhR knockout subsequently led to decreased SCC formation in a UVB mouse model [41]. Further, in estrogen receptor (ER)-positive breast cancer, AhR signaling increased antiapoptotic x-linked inhibitor of apoptosis (XIAP) and superoxide dismutase type 1 in two cell lines, leading to apoptotic resistance [42]. In pancreatic cancer, the AhR ligand kynurenine increased expression of multiple anti-apoptotic proteins, including XIAP and B-cell lymphoma 2 (Bcl-2), while decreasing the pro-apoptotic protein bax, which was abrogated with the addition of the AhR inhibitor CH-223191 [43]. Taken together, these studies show that activation of AhR enhances tumor cells’ ability to resist apoptosis.

### 2.4. Angiogenesis

Angiogenesis is critical to providing oxygen and nutrients to a proliferating tumor and impacts the propensity for metastasis. AhR activation has been shown to promote angiogenesis through a variety of mechanisms. Activated AhR-ARNT heterodimers interact with hypoxia-inducible factor (HIF)-1α to increase expression of interleukin-8 and vascular endothelial growth factor (VEGF), downregulate expression of transforming growth factor beta, and promote new vessel formation [44,45]. This AhR-mediated angiogenesis appears to be dependent upon VEGF: while AhR null mice had impaired angiogenesis, the phenotype was rescued with the addition of VEGF [45].

## 3. Role in Innate Immunity

AhR has been implicated in the differentiation and function of a variety of both innate and adaptive immune cells. In this section, we review the known impact of AhR on various immune subtypes (Figure 2), as well as the limitations of current findings.

### 3.1. Dendritic Cells

Dendritic cells (DCs) and macrophages are antigen-presenting cells (APCs) that mediate adaptive immune responses through processing and presentation of foreign antigens. The impact of AhR signaling on DC function is complex and context dependent. AhR expression is important in monocyte-derived type 1 DC differentiation, which plays an important role in the anti-tumor response [46,47]. However, AhR has also been found to regulate differentiation of tolerogenic DCs in mice, defined by decreased co-stimulatory signals, immature phenotype, and expression of inhibitory molecules and cytokines [48]. Tolerogenic DCs have been shown to induce immunosuppressive regulatory T cells (T_regs_) through an AhR-dependent pathway mediated by tryptophan metabolites and nuclear coactivator 7 [49]. The dichotomous impact of AhR on dendritic cells underscores the need for further studies to understand its true role in DC subsets and its impact on tumor immunity.

### 3.2. Macrophages

Macrophages are APCs that are thought to span a continuum from a more inflammatory M1 phenotype associated with nitric oxide expression to an anti-inflammatory M2 phenotype associated with arginine expression [50]. M1 macrophages are thought to be more anti-tumorigenic, while M2 macrophages suppress cytotoxic T cell function, thereby impairing immune surveillance [51]. AhR is critical in macrophage polarization, as demonstrated by AhR knockout in M1-polarized macrophages. This leads to production of more inflammatory cytokines, but significantly decreased phagocytic capacity, rendering mice more susceptible to infection. Interestingly, AhR knockout in M2-polarized macrophages decreases immunoregulatory IL-10 production, but increases arginase activity, thereby heightening their immunosuppressive capabilities [52]. Macrophages found within the TME are known as tumor-associated macrophages (TAMs), and their polarization state and functionality are similarly influenced by AhR activation, leading to enhanced immunosuppression. In melanoma, increased expression of IDO1 or tryptophan 2,3-dioxygenase (TDO), another enzyme involved in metabolizing tryptophan to AhR ligands, increases the abundance of TAMs and polarizes to a more M2-like immunosuppressive phenotype [17]. In a pancreatic cancer mouse model, TAMs demonstrated high AhR activity, and a reduction in AhR deficiency activity in TAMs led to a more inflammatory phenotype associated with control of tumor growth [19].

In addition to influencing the overall polarization state of TAMs, T cell–macrophage crosstalk is critical to anti-tumor immunity and regulated by AhR activation. In glioblastoma, macrophage-derived kynurenine suppressed T cell-mediated immunity, and macrophage-specific deletion of AhR in an intracranial glioblastoma mouse model reduced tumor growth [53]. Kynurenine-mediated AhR activation led to increased C-C motif chemokine receptor 2 (CCR2) activation and TAM recruitment, as well as recruitment of ectonucleotidase CD39, which promoted CD8^+^ T cell dysfunction [53]. In line with this, AhR activation promoted expression of immunosuppressive programmed death-ligand 1 (PDL1) in TAMs, and AhR inhibition mildly decreased the tumor size and improved the CD8 to T_reg_ ratio in a separate glioblastoma model [54]. In pancreatic cancer, macrophage-specific deletion of AhR or AhR inhibition impaired tumor growth, improved the efficiency of checkpoint blockade therapy, and increased infiltration of CD8^+^ T cells [19]. In general, AhR activation appears to polarize to an immunosuppressive macrophage phenotype, while dampening T cell function.

### 3.3. Myeloid-Derived Suppressor Cells

Myeloid-derived suppressor cells (MDSCs), a heterogenous group of neutrophils and monocytes with potent immunosuppressive activity, are known to impair the anti-tumor immune response [55]. Evidence of the role of AhR in MDSC tumor immunity is limited. There is some evidence that intraperitoneal exposure to the prototypical ligand TCDD may increase MDSCs within the peritoneal cavity and spleen among wild-type mice [56]. However, it is possible that this finding was unrelated to actual activation by TCDD and may instead be related to dysbiosis, as TCDD was shown to alter the gut microbiome and bacterial tryptophan metabolism [57]. Additionally, the non-microbial-associated indole-3-proprionic acid, an AhRL produced from dietary tryptophan, was also shown to induce polymorphonuclear MDSC differentiation in vitro [58]. Given these limited results, further studies are needed to understand the role of AhR in MDSC differentiation and function.

### 3.4. Natural Killer Cells and Innate Lymphoid Cells

Natural killer (NK) cells are cytotoxic innate immune cells of lymphoid origin that are integral in viral and anti-tumor responses [59]. A correlation between NK infiltration and overall survival has been seen in multiple cancer types [60], and AhR activation appears to potentiate NK cell cytolytic activity. Cytokine stimulation with interleukin (IL) 2, IL15, or IL12 induced AhR expression in NK cells of mice in vitro, while knockout of AhR in NK cells reduced their cytolytic activity and ability to control lymphoma formation [61]. In addition to improving cytolytic activity, AhR activation has been shown to promote NK cell migration. Murine Ahr^−/−^ NK cells had reduced capacity to migrate, associated with decreased *ankyrin repeat and SOCS box protein 2* (*Asb2*) gene expression. Similarly, *Asb2* knockdown in human NK cells demonstrated reduced migratory capacity [62]. Taken together, these studies suggest that AhR activation appears to be critical for proper NK cell function.

Innate lymphoid cells (ILCs) are innate immune cells derived from common lymphoid progenitors. ILCs are primarily tissue resident and respond to tissue damage via secretion of cytokines to mediate both innate and adaptive immunity. The three general families of ILCs (types 1, 2, and 3) differ in their principal transcription factor and cytokine production profile. Type 1 ILCs are generally thought to be pro-inflammatory and, therefore, anti-tumor; however, there is conflicting evidence on the overall role of type 2 ILCs (ILC2) and their impact on tumorigenesis. For instance, ILC2s have been shown to both suppress NK cell-mediated tumor cytotoxicity via interleukin-33 in melanoma [63] and to promote MDSC differentiation and tumor establishment [64,65]. However, high ILC2 infiltration has also been associated with improved prognosis in melanoma [66] and pancreatic ductal adenocarcinoma (PDAC) [67]. Further studies are needed to determine the role of ILC2s in tumor immunity given the context-dependent results to date.

In contrast, type 3 ILCs (ILC3), driven by the retinoic acid receptor-related orphan receptor gamma t (RORγt) transcription factor, are highly sensitive to AhR activation. ILC3s are generally seen as pro-tumorigenic, as suggested by studies in both hepatocellular carcinoma [68] and colorectal cancer [69], in part due to their ability to induce tumor cell signal transducer and activator of transcription 3 (STAT3) activation and cellular proliferation [69,70]. In breast cancer, ILC3s are associated with lymph node metastasis, and in mice, depletion of ILC3s decreased lymph node metastasis [71]. AhR regulates the differentiation and proliferation of ILC3s and is necessary for interleukin-22 (IL22) production within ILC3s [72,73]. IL22 has been shown to promote cancer formation in both the colon and pancreas. AhR plays a critical role in driving the balance of ILCs, with activation of AhR shown to suppress ILC2s and promote ILC3s [74,75]. Consistent with this, murine AhR knockouts have more ILC2s in the gut compared to their wild-type counterparts [74]. Similarly, ILC3s and IL22 expression was decreased in Ahr^−/−^ and Ahr^fl/fl^Rorc-cre mice, while the constitutively expressed knock-in Ahr^+/dCAIR^ demonstrated increased ILC3s and IL22 [74,75].

### 3.5. Role in Adaptive Immunity

#### 3.5.1. CD8^+^ T Cells

There is conflicting evidence on the role of AhR in CD8^+^ T cell subsets, where activation has been shown to induce both pro-tumorigenic and anti-tumorigenic phenotypes. AhR activation promotes T cell exhaustion and reduces effector T cell function, but at the same time promotes formation of tissue-resident memory (T_RM_)-like cells associated with anti-tumor responses and disease control. In a murine model of influenza A infection, TCDD-treated mice showed reduced CD8^+^ effector T cells in lungs and lymph nodes in an AhR-dependent manner [76]. In tumors, kynurenine activated AhR and led to upregulated programmed cell death protein 1 (PD-1) expression on CD8^+^ T cells, promoting T cell exhaustion. Blockade with the AhR antagonist 3′,4′-dimethoxyflavone impaired tumor growth and improved survival in B16GF10 melanoma [77]. Similarly, in an orthotopic model of oral SCC, AhR-knockout in oral SCC cells led to smaller tumors and increased activated T cells, while controls had increased expression of exhaustion markers and the checkpoint inhibitors PD-1, cytotoxic T-lymphocyte protein-4 (CTLA-4), and lymphocyte activation gene 3 (Lag3) [78].

T_RM_ cells are a subset of memory T cells that remain within the tissue and are found predominantly in mucosal linings and skin at common sites of pathogen exposure. T_RM_ are CD8^+^ T cells often identified by co-expression of cluster of differentiation (CD) 69 and CD103 [79]. Recently, CD69^+^CD103^+^ T_RM_-like T cells have been found in solid tumors, where increased infiltration is associated with improved outcomes and survival [80,81]. For instance, in a melanoma model, T_RM_-like cells promoted spontaneous disease control, preventing tumor outgrowth in mice with occult melanoma [82]. Similarly, in a preclinical head and neck cancer model, induction of local T_RM_-like cells inhibited tumor growth [83] in lung cancer, wherein T_RM_-like cells improved survival [81]. T_RM_-like cells had a T cytotoxic (Tc) type 17 (Tc17)-like phenotype with high AhR and phospho-STAT3 expression [84].

An additional anti-tumorigenic role for AhR activation is seen with Tc22 cells, CD8^+^IL22^+^ cytolytic T cells that are dependent upon AhR and IL6 for differentiation and function [85]. In ovarian cancer, Tc22 cells’ and IL22 production was associated with improved survival [85]. Since AhR can have both pro- and anti-tumorigenic roles within the CD8^+^ T cell compartment, further studies are needed to better understand the context dependence of AhR activation.

#### 3.5.2. CD4 T Cells

AhR plays a vital role in the differentiation of regulatory T cells (T_regs_). T_regs_ expressing the transcription factor forkhead box P3 (Foxp3) were induced by the ligand TCDD, while activation of AhR by the endogenous ligand FICZ inhibited T_reg_ differentiation and promoted T helper (Th)17 differentiation, suggesting a ligand-specific role in T_reg_ development [86]. Similarly, type 1 regulatory T cells (Tr1), a subset of immunosuppressive CD4^+^ T cells that inhibit antigen-specific T cell responses but differ from traditional T_regs_ through their lack of Foxp3 expression, relied upon AhR for differentiation and maintenance of activation [87,88]. CD39, ectonucleoside triphosphate diphosphohydrolase-1, drives a shift from a pro- to an anti-inflammatory milieu by increasing production of adenosine, which is critical in regulating the immunosuppressive function of T_regs_ [89]. AhR has been shown to increase expression of CD39 via the STAT3 signaling pathway and, therefore, immunosuppressive function [88]. Together, these studies suggest that ligand-specific AhR activation drives T_reg_ differentiation and, in general, promotes immune suppression.

In addition to its direct action on T_regs_, AhR activity has been implicated in crosstalk among T_regs_ and both macrophages and dendritic cells (DCs). IDO1 and TDO2 are key enzymes in the catabolism of tryptophan into kynurenine: IDO1 and TDO2 expression has been shown to be immunosuppressive through action on MDSCs and T cells [90,91], and has been associated with resistance to ICI therapy in preclinical models [90,92]. Campesato et al. found that tumors with IDO1 and TDO2 generate kynurenine and activate AhR to drive generation of T_regs_ and tolerogenic TAMs. Selective AhR blockade in tumors with high expression of IDO1 and TDO2 reversed T_reg_- and TAM-mediated immunosuppression and improved the efficacy of PD-1 blockade [17]. Through an additional mechanism, DCs treated with an AhR agonist induced T_reg_ formation via IL-2 and IL-10, acting upon the negative immune regulator B7-H4 [93].

AhR regulates the differentiation and function of other CD4^+^ T helper cells. Th22 and Th17 cells, for example, are driven by the transcription factor RORγt, but Th22 cells are distinct from Th17 cells in that they produce IL22 but not IL17 [94]. The role of AhR activation on Th17 differentiation appears to be ligand-specific. FICZ has been shown to induce Th17 differentiation [95], whereas TCDD led to expansion of T_regs_ and decreased Th17 differentiation [86]. However, AhR activation appears to regulate Th22 differentiation [86,95,96] by preferentially differentiating to IL22-producing cells over IL17-producing subtypes [95]. The role of IL22 and, therefore, AhR-dependent T helper cell differentiation in cancer progression, is context dependent, with both pro-tumor and anti-tumor roles described [97,98]. IL22 signaling has been implicated in increasing stemness, tumor proliferation, tumor migration and invasion, and anti-apoptotic resistance in multiple cancer types [98]. However, in colitis-associated colorectal cancer (CRC), AhR has been shown to serve a protective role against CRC, which is thought in part to be due to the role of IL22 in tissue repair and regeneration in the gastrointestinal tract [99,100]. In a preclinical model utilizing Villin^cre^AhR^fl/fl^ mice, azoxymethane alone led to the development of CRC, an effect not seen in wild-type mice [101].

## 4. Additional Mechanisms of Tumor Immune Evasion

AhR signaling has also been implicated in other mechanisms of tumor immune evasion, including upregulation of tryptophan metabolism, induction of checkpoint inhibitors, and induction of stemness and chemoresistance within tumor cells (Figure 1).

### 4.1. Induction of Immune Checkpoints and ICI Resistance

Another common mechanism of tumor immune evasion involves upregulation of immune checkpoints, such as PD-1 and its ligand programmed death-ligand 1 (PD-L1), or CTLA4. AhR signaling has been shown to impact immune checkpoint expression directly on tumor cells and within tumor-infiltrating lymphocytes. The kynurenine-AhR pathway directly upregulates PD-1 expression on CD8^+^ T cells, while interferon gamma (IFNγ) expression from activated CD8^+^ T cells drives further kynurenine production from tumor repopulating cells, leading to paracrine deactivation [77]. Similarly, the AhRL benzo(a)pyrene (BaP) induced PD-L1 expression on lung epithelial cells and promoted lung cancer progression, while anti-PD-L1 antibodies or AhR deficiency suppressed BaP-induced lung cancer progression [102]. Among patients treated with pembrolizumab in a lung cancer cohort, high AhR expression was associated with partial response or stable disease, whereas low AhR expression was associated with disease progression [102]. Similarly, the immune checkpoint molecules CTLA4 and Lag3 are known to be upregulated with AhR signaling [78].

### 4.2. Stemness and Chemoresistance

AhR has been shown to direct hematopoietic progenitor cell expansion and differentiation [103,104]. AhR is critical for normal function of hematopoietic progenitor cell populations, and normal function of these populations is disrupted in AhR knockout mice [105,106], suggesting a role for AhR signaling in stem-related cell function.

Cancer stem cells drive tumor initiation and resistance to cancer therapy and have been detected in a wide variety of solid tumors [107]. Stemness within tumors is characterized by a reduced level of tumor differentiation and increased self-renewal capacity [108]. AhR has been shown to influence expression of stem-related genes in multiple cancer types. In triple-negative breast cancer, AhR activation led to upregulation of stem-related gene expression [37]. In spheroids developed from human colon cancer samples, stemness characteristics were dependent upon AhR expression [109].

A hallmark of cancer stemness is resistance to chemotherapy. One mechanism by which AhR supports chemoresistance is through the upregulation of the transporter ATP-binding cassette super-family G member 2 (ABCG2), which has been shown to export chemotherapy drugs in multiple cancers, including breast [110], choriocarcinoma [111], and nasopharyngeal and lung [112]. Similarly, aldehyde dehydrogenase (ALDH) contributes to drug export from tumor cells, through upregulation of ATP-binding cassette super-family B member 1 (ABCB1) [113], and its expression is associated with both chemo- and radio-resistance [114,115]. AhR activation has been shown to upregulate ALDH in oral cancer [116] and breast cancer [117], while in triple-negative breast cancer, AhR inhibition reduced chemoresistance in ALDH^high^ tumors [37]. Similarly, inactivation of AhR in cancer stem cells in a breast cancer model sensitized tumors to the chemotherapy doxorubicin [117].

However, the role of AhR activation in stemness and chemoresistance can be tumor dependent. For instance, in acute myeloid leukemia, AhR ligands impaired cell growth and suppressed self-renewal [118]. Additionally, selective AhR agonists with mild activating properties, and certain endogenous ligands, can be anti-tumorigenic [119,120]. This context dependence may be due to tissue-specific and ligand-specific effects of AhR, as these can differentially recruit AhR cofactors depending on the tissues and ligands [121,122].

## 5. Therapeutic Potential and Future Directions

### 5.1. Direct Targeting of AhR

Given the importance of AhR in pathologic immune cell polarization, its role in carcinogenesis, and impact on tumor immune evasion, targeting AhR offers an exciting opportunity for cancer therapy. Directly targeting AhR has shown some promise in preclinical models. Campesato et al. showed decreased tumor growth and controlled regulatory T cell formation with the AhR inhibitor CH-223191 and the AhR antagonist Kyn-101 from Ikena Oncology [17]. The AhR inhibitor HP163, developed by Hercules Pharmaceuticals, reduced tumor growth in oral, breast, and skin orthotopic tumor models, and was shown to decrease immunosuppressive CD11b^+^ cells in the draining lymph nodes of oral squamous cell cancer [123]. An additional AhR inhibitor, BAY-218 (from Bayer AG, Leverkusen, Germany), stimulated pro-inflammatory monocyte and T cell responses in vitro and anti-tumor responses in vivo in tumor models using CT16 and B16-OVA cell lines [124]. Phase I clinical trials are underway studying two oral AhR inhibitors, BAY2416964 and IK-175, in patients with advanced solid tumors unresponsive to prior treatments [125,126]. Interim results showed that of the 67 patients treated with BAY2416964, 32.8% demonstrated stable disease [127].

Similarly, as upregulation of PDL1 and other inhibitory checkpoint molecules is a common mechanism of acquired resistance to immunotherapy that is enhanced by AhR-dependent signaling [77,78,102], a strategy of AhR inhibition in combination with ICI therapy makes logical sense. Phase Ib clinical trials are underway investigating BAY2416964 and pembrolizumab in patients with advanced solid tumors [128], as well as IK-175 in combination with nivolumab in primary PD-1-inhibitor-resistant metastatic or locally incurable recurrent head and neck squamous cell carcinoma [129].

However, blockade of the AhR signaling pathway requires careful consideration, as AhR agonism can be associated with anti-tumor effects in certain contexts (Figure 2) [119,120]. Additionally, AhR activation plays a number of physiologic roles, and its deletion has been shown to induce several detrimental phenotypes in AhR-null mice, including impairment of vascular differentiation, immune abnormalities, and liver abnormalities [106,130,131]. Interim analysis of the phase I trial of systemic AhR inhibition with BAY2416964 showed that the compound was generally well tolerated, with no dose-limiting toxicities, but 12.5% of patients experienced grade 3 treatment-emergent adverse events (TEAEs), while the remainder of adverse events were grade 1 or 2 [127]. Similarly, when considering ICI therapy, although the rates of adverse events vary with the specific ICI agent used, complications are common. Severe immune-related adverse events range from 0.5% to 13% for single-agent therapy, while up to 43% of patients on combination ICI therapy discontinued therapy due to adverse events [132,133]. While systemic AhR blockade may be an effective method for AhR modulation, further clinical trials are needed to determine safety and efficacy, both for AhR inhibition alone and in combination with immune checkpoint blockade.

### 5.2. Kynurenine Depletion and Dual Blockade of IL4I1 and IDO/TDO

Upregulation of tryptophan catabolism by IDO1 and TDO2 is a well-described mechanism of cancer immune evasion. IDO1 and TDO2 expression are associated with numerous tolerogenic immune cells, including suppression of effector T cells and infiltration of MDSCs and T_regs_ in multiple cancer types [134]. AhR activation is believed to be the primary mechanism linking tryptophan metabolism to immune suppression in the TME. One strategy for modulation of AhR signaling in cancer treatment entails eliminating the pool of immunosuppressive AhR ligands, either through inhibition of tryptophan catabolism or by promoting clearance of the ligands. A preclinical study of PEGylated kynureninase, an enzyme that degrades kynurenine into an immunologically inert byproduct, thereby decreasing the pool of immunosuppressive AhR ligands, was associated with decreased tumor growth and increased CD8^+^ T cell infiltration in breast, melanoma, and colon cancer cell lines when combined with ICI therapy [18].

Direct inhibition of IDO1 and TDO2 is an alternate strategy to decrease kynurenine production and target tumor immune evasion. Despite the promise of IDO/TDO blockade in preclinical studies, a phase III clinical trial in unresectable melanoma found no difference in survival between IDO1 blockade with ICI therapy when compared to ICI therapy alone [20]. Early clinical trials revealed that complete blockade of tryptophan catabolism with IDO and/or TDO inhibition could lead to non-specific activation of AhR, as well as non-specific activation of mammalian target of rapamycin (mTOR) signaling, which can induce cell growth and proliferation [135]. These non-specific actions appear to limit the efficacy of IDO1 inhibition. Additionally, the reliance of IDO on tryptophan metabolism is variable, with some tumors having very little expression. A study of advanced melanoma revealed that only 39.5% of patients had tumors with elevated IDO expression, and only 9.3% expressed both IDO1 and PDL1 [136]. In a preclinical study of melanoma, IDO inhibition in IDO-expressing B16 melanoma, but not wild-type melanoma, decreased the tumor size, while direct AhR inhibition decreased the tumor size in both IDO- and TDO-expressing tumors [17]. There may be a role for molecular profiling to target tryptophan catabolism or AhR in tumors with high IDO and TDO expression, but this requires further study.

Recently, IL4I1 was shown to catalyze a secondary tryptophan catabolic pathway to the AhRLs kynurenic acid and other indoles [21]. IL4I1 is upregulated in multiple tumor types, associated with poor prognosis, and is more closely associated with AhR activation than IDO expression. Together, these data suggest that IDO or TDO inhibition may be inadequate for suppression of AhR-mediated immunosuppression, and that IL4I1 expression could in part explain the failure of combination ICI and IDO1 inhibition observed in previous studies. Combination of IDO and/or TDO inhibition with IL4I1 blockade is a promising therapeutic strategy that deserves additional investigation to evaluate its safety and efficacy.

### 5.3. Modulation of the Microbiome and Diet

The microbiota of the gastrointestinal tract has been shown to systemically impact innate and adaptive immunity, with both pro- and anti-tumor effects through the regulation of signaling pathways, growth factor production, and inflammation [137,138,139]. As such, gut microbiota play a large role in responsiveness to chemotherapy, radiotherapy, and immunotherapy [138]. For instance, modulation of the microbiome through fecal microbiota transplant (FMT) from ICI responders improved the efficacy of PD-1 blockade, whereas FMT from non-responders showed no improved response in mouse models of melanoma [140,141]. This effect of the microbiome regulating the response to ICI therapy was further demonstrated in a phase II single-arm trial, where FMT partially reversed checkpoint inhibitor resistance in PD-1 refractory melanoma [142].

Both diet and microbiota are significant sources of indoles and metabolites with AhR activity [143]. These AhR metabolites have broad effects on immune tolerance and response [3], carcinogenesis [2], and epithelial integrity within the GI tract [144]. Decreasing AhR ligand production from the gut microbiota is an additional strategy for indirect AhR modulation in cancer therapy. Hezaveh et al. found that in a pancreatic cancer model, indoles from the gut microbiome drove alternative TAM polarization and tumor growth, and depletion of indole-producing bacteria or dietary tryptophan blocked this effect [19]. Interestingly, in a preclinical melanoma model, Bender et al. found that the probiotic-released AhR ligand indole-3-aldehyde promoted anti-tumor immunity via interferon-γ-producing CD8^+^ T cells and improved ICI efficacy [145]. Further studies on the context-dependent and ligand-specific effects of microbial metabolites on AhR signaling are needed.

## 6. Conclusions

AhR has broadly been shown to directly impact carcinogenesis through its role in cellular proliferation, EMT, angiogenesis, stemness, and induction of chemoresistance. Its impact on tumor immunity is more complicated; in general, it is context- and cell-type-specific, with both pro- and anti-tumorigenic roles. Further insights are needed to classify the role of AhR signaling in immune cell polarization, as this appears to be the primary way AhR signaling impacts the immune microenvironment. Given the contrasting roles of AhR activation, and the potential consequences of systemic AhR blockade, additional factors need to be considered when developing strategies to target AhR in cancer therapy. While modulation of AhR signaling through inhibition of tryptophan metabolism offers promise, the lack of a survival benefit in a phase III clinical trial suggests alternative strategies are needed, such as molecular profiling to target high-IDO-expressing tumors or blockade of IL4I1. Similarly, further studies of direct AhR inhibition by modulation of the microbiota could provide a novel means for boosting anti-tumor immunity.

## Figures and Tables

**Figure 1 cancers-16-00472-f001:**
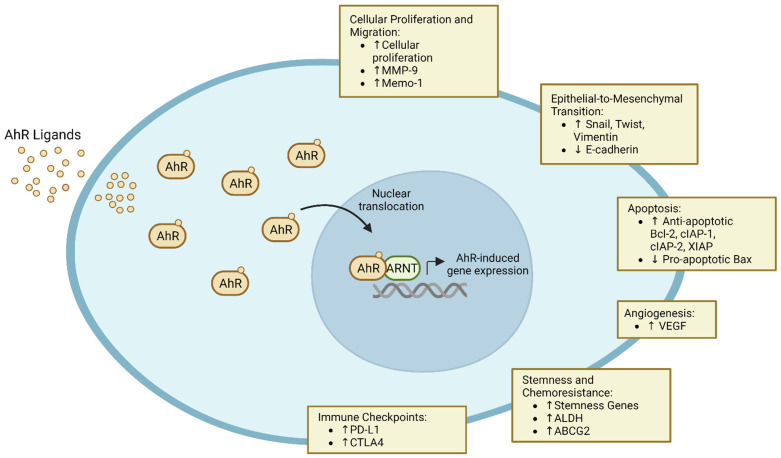
Tumor-associated AhR activity is often increased compared to adjacent normal tissue. The AhR is a cytoplasmic receptor responsive to a variety of endogenous and exogenous ligands that translocates to the nucleus upon activation. Elevated AhR activity leads to a variety of downstream tumor-intrinsic effects that impact tumor proliferation and therapeutic resistance, including modulation of cellular proliferation and migration, epithelial-to-mesenchymal transition, apoptosis, angiogenesis, stemness, and expression of immune checkpoints. Abbreviations: MMP-9 (matrix metalloproteinase-9), Memo-1 (mediator of cell motility 1), Bcl-2 (B-cell lymphoma 2), cIAP (cellular inhibitor of apoptosis protein), XIAP (x-linked inhibitor of apoptosis protein), VEGF (vascular endothelial growth factor), ALDH (aldehyde dehydrogenase), ABCG2 (ATP-binding cassette super-family G member 2), PD-L1 (programmed death-ligand 1), and CTLA4 (cytotoxic T-lymphocyte-associated protein 4).

**Figure 2 cancers-16-00472-f002:**
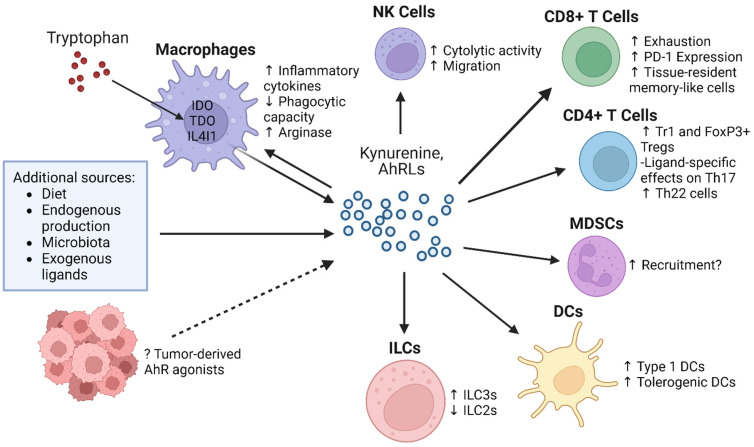
Aryl hydrocarbon receptor ligands’ impact on tumor immunity. Summarized here is the impact of AhR activation on macrophages, NK cells, CD8^+^ and CD4^+^ T cells, MDSCs, DCs, and ILCs. There are a variety of sources of AhR ligands within the tumor microenvironment, and activation of AhR in immune cells can lead to pro- or anti-tumorigenic effects. For instance, AhR activation in macrophages has been shown to increase inflammatory cytokine secretion but decrease phagocytic capacity and increase immunosuppressive arginase expression. In CD8^+^ T cells, AhR activation leads to increased exhaustion markers and immune checkpoint expression, as well as increased tissue-resident memory-like T cells. CD4^+^ T cells are subject to ligand-specific effects, but AhR activation generally increases FoxP3+ and Tr1 T_regs_ as well as Th22 cells. Additionally, AhR activation promotes both type 1 DCs and tolerogenic DCs. Abbreviations: NK (natural killer), IDO (indoleamine 2,3-dioxygenase), TDO (tryptophan 2,3-dioxygenase), IL4I1 (interleukin 4-induced 1), MDSC (myeloid-derived suppressor cell), DC (dendritic cell), Tr1 (Type 1 regulatory T cell), Treg (regulatory T cell), ILC (innate lymphoid cell), and PD-1 (programmed cell death receptor-1).

**Table 1 cancers-16-00472-t001:** Common AhR agonists and antagonists listed by source [2,3,4].

Source	Ligand
**Agonists**	
**Endogenous**	*Tryptophan Metabolites:* - Kynurenic acid - Kynurenine - 6-formylindolo[3,2b]carbazole (FICZ) - Indoxyl sulfate
	*Heme-Derived:* - Bilirubin - Biliverdin
	*Arachidonic Acid Metabolites:* - Lipoxin 4 - Prostaglandin PGG2 - Hydroxyeicosatrienoic acid
**Dietary**	*Indoles:* - Indole-3-carbinol - 3,3′-diindoylmethane - Indolo[3,2b]carbazole
	*Flavonoids:* - Quercetin - Galangin
**Microbiota**	- Indirubin - Indol-3-acetic acid - Indole-3-aldehyde - Tryptamine - 1,4-dihydroxy-2-napthoic acid
**Xenobiotic**	*Halogenated Aromatic Hydrocarbons:* - 2,3,7,8-tetrachlorodibenzo-*p*-dioxin - Benzo[*a*]pyrene - Benzanthracenes - Benzoflavones - Biphenyls - Polyaromatic hydrocarbons
	*Other:* - Omeprazole - Tranilast - Leflutamide
**Antagonists**	
**Dietary**	- Resveratrol
**Xenobiotic**	- CH-223191 - StemRegenin 1 - GNF352

## Data Availability

The data can be shared up on request.

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
