# Peer review of "The Aryl Hydrocarbon Receptor: Impact on the Tumor Immune Microenvironment and Modulation as a Potential Therapy"

_cancers, 2024, doi:10.3390/cancers16030472_

Round 1

Reviewer 1 Report

Comments and Suggestions for Authors

Specific comments:

1.       AhR is cytoplasmic. It translocates into nucleus upon activation.

2.       Provide information on known AhR ligands (agonists and antagonists) in a table format.

3.       MMPs -  matrix metallopeptidases or matrix metalloproteinases?

4.       Cellular Proliferation and Migration section.

a.       Considering the main focus of this review on TME, the authors should specify if AhR agonist displayed such effect in the described here breast, gastric and colon cancers.

b.       Abbreviations should be defined upon their first mentioning in the manuscript text, such as CRC, etc.  

c.       It is unclear how the loss of cell-cell contact promotes cellular migration and invasion.

5.       Is AhR expression downregulated or upregulated in different types of cancer? How is its upregulation or downregulation associated with cancer progression and median survival?

6.       Mark nuclear translocation in Figure 1.

7.       Apoptosis section. BAX, bcl-2-like protein 4.

8.       The statement on what kind of further work is needed should be eliminated from Figure 2 legend. Add PD-1 expression on CD8+ T cells. Mark a ligand-specific effect of AhR activation on Treg cells.

9.       Specify what do you mean under terms “immunoregulatory DC” and “pathologically activated neutrophils”.

10.   A prototypical ligand TCDD may increase MDSCs within the peritoneal cavity and spleen. In what case? When injected intraperitoneally into the mice bearing tumor (and what type)? Where an AhRL induced MDSC differentiation?

11.   Cytokine stimulation induced AhR expression in NK cells of mice. Does this mean that normally NK cells do not express AhR? What types of cytokines induce such expression? What cytokines suppress it? Provide a full definition for Asb2.

12.   The statement “Importantly, no data exist on the role of AhR in the function of either type 1 or type 2 ILCs” contradicts with following statement “Activation of AhR shown to suppress ILC2s and promote ILC3s”. What is AhR activation effect on ILC2? Provide corresponding references. See also Figure 2.

13.   Define mentioned here CD8+ T cell subsets.

14.   It is unclear if AhR activation inhibits Th17 response.

15.   Additional Mechanisms of Tumor Immune Evasion. It is in Fig. 1 instead of mentioned here Fig. 2. Although tryptophan metabolism is mentioned in Fig. 2.

16.   What is the mechanism of ALDH-mediated drug export from tumor cells?

17.   Modulation of the Microbiome and Diet. Microbiota can influence cancer immunity in both positive and negative ways. Indirect AhR inhibition by modulation of the microbiota.

Author Response

Thank you very much for your time and effort in reviewing our article “The Aryl Hydrocarbon Receptor: Impact on the Tumor Immune Microenvironment and Modulations as a Potential Therapy” and its submission to your journal Cancers. Please see our response and changes to your insightful comments below.

Reviewer 1:

  1. AhR is cytoplasmic. It translocates into nucleus upon activation.

Thank you for noting this error, which has been updated in the text.

  1. Provide information on known AhR ligands (agonists and antagonists) in a table format.

A table detailing known AhR ligands, as well as references to prior in-depth reviews with additional information, have been added to the text.

  1. MMPs -  matrix metallopeptidases or matrix metalloproteinases?

Thank you for your comment. All references were adjusted to matric metalloproteinases and are now consistent.

  1. Cellular Proliferation and Migration section.
    1. Considering the main focus of this review on TME, the authors should specify if AhR agonist displayed such effect in the described here breast, gastric and colon cancers.

Thank you. The cancer type under investigation has been included in the cellular proliferation and migration section.

  1. Abbreviations should be defined upon their first mentioning in the manuscript text, such as CRC, etc. 

Definitions for multiple abbreviations were added to the manuscript text. Thank you.

  1. It is unclear how the loss of cell-cell contact promotes cellular migration and invasion.

A description and reference detailing in further depth how the loss of cell-cell contact promotes migration and invasion has been added to the text.

  1. Is AhR expression downregulated or upregulated in different types of cancer? How is its upregulation or downregulation associated with cancer progression and median survival?

AhR expression is chronically active and upregulated in a number of cancer types, detailed in the first paragraph of the “Role in Carcinogenesis” section. Additional information detailing the link between upregulation and worse survival outcomes, as well as an in depth review of AHR expression and survival in cancer, has been added. Thank you.

  1. Mark nuclear translocation in Figure 1.

Nuclear translocation noted in figure 1.

  1. Apoptosis section. BAX, bcl-2-like protein 4.

These abbreviations have been defined in the text. Thank you.

  1. The statement on what kind of further work is needed should be eliminated from Figure 2 legend. Add PD-1 expression on CD8+ T cells. Mark a ligand-specific effect of AhR activation on Treg cells.

Described updates to figure 2 have been included. Thank you.

  1. Specify what do you mean under terms “immunoregulatory DC” and “pathologically activated neutrophils”.

The term “immunoregulatory” has been updated to the better descriptor “tolerogenic.” A description of tolerogenic DCs, including decreased co-stimulatory signals, immature phenotype, and decreased expression of inhibitory molecules and production of anti-inflammatory cytokines, was included. Additionally, the description of MDSCs was changed to a heterogenous of group of immunosuppressive monocytes and neutrophils, and the vague term “pathologically activated” neutrophils was removed. Thank you.

  1. A prototypical ligand TCDD may increase MDSCs within the peritoneal cavity and spleen. In what case? When injected intraperitoneally into the mice bearing tumor (and what type)? Where an AhRL induced MDSC differentiation?

Clarification of the limited evidence surrounding AhR activity on MDSC differentiation and function was added. Intraperitoneal exposure to TCDD increased MDSCs both within the peritoneal cavity and spleen in wild-type mice. The AhRL indole-3-proprionic acid induced polymorphoneuclear MDSC differentiation  in vitro

  1. Cytokine stimulation induced AhR expression in NK cells of mice. Does this mean that normally NK cells do not express AhR? What types of cytokines induce such expression? What cytokines suppress it? Provide a full definition for Asb2.

NK cells normally express AhR at low levels. Cytokines that induced AhR expression in NK cell in vitro included IL-2, IL-15, or IL-12. This information, as well as a full definition of Asb2, was added to the text.

  1. The statement “Importantly, no data exist on the role of AhR in the function of either type 1 or type 2 ILCs” contradicts with following statement “Activation of AhR shown to suppress ILC2s and promote ILC3s”. What is AhR activation effect on ILC2? Provide corresponding references. See also Figure 2.

This contradiction was removed from the text, and the citations noting the impact of AhR activation on ILC2 and ILC3 balance have been updated.

  1. Define mentioned here CD8+ T cell subsets.

Clarifying statements have been added to the CD8 section of the text, and the tissue-resident memory like cell subset of CD8 T cells are defined, as are the cytolytic CD8+IL22+ T cell subset, TC22 cells.

  1. It is unclear if AhR activation inhibits Th17 response.

The impact of AhR activation on Th17 response appears to be ligand-dependent, as with regulatory T cell differentiation. This was made clear in the text and additional references were added.

  1. Additional Mechanisms of Tumor Immune Evasion. It is in Fig. 1 instead of mentioned here Fig. 2. Although tryptophan metabolism is mentioned in Fig. 2.

Thank you for noting this error, which has been corrected.

  1. What is the mechanism of ALDH-mediated drug export from tumor cells?

ALDH has been associated with drug resistance through increasing expression of stemness genes, as well as drug export through upregulation of ALDH-dependent drug-efflux pump ABCB1 (ATP binding cassette subfamily B member 1). This has been included in the text. Thank you.

  1. Modulation of the Microbiome and Diet. Microbiota can influence cancer immunity in both positive and negative ways. Indirect AhR inhibition by modulation of the microbiota.

It is important to acknowledge both the pro- and anti-tumor effects of the microbiome on tumor progression. This has been added to the text.

Reviewer 2 Report

Comments and Suggestions for Authors

The authors have addressed a timely topic and appear to have drawn on all the relevant literature.  The paper reads well and the authors successfully pointed out the current inconsistencies and challenges in the field.  Where I feel the paper still needs work is in the therapeutic potential and future directions section.  As presently written, this section just describes the current ongoing trials/work; what is missing are some insightful new directions, connection, experiments, and therapeutic modalities to consider.  Without these, the MS is somewhat flat.

Author Response

Reviewer 2:

The authors have addressed a timely topic and appear to have drawn on all the relevant literature.  The paper reads well and the authors successfully pointed out the current inconsistencies and challenges in the field.  Where I feel the paper still needs work is in the therapeutic potential and future directions section.  As presently written, this section just describes the current ongoing trials/work; what is missing are some insightful new directions, connection, experiments, and therapeutic modalities to consider.  Without these, the MS is somewhat flat.

Clarifying statements were added to therapeutic potential and future directions section to attempt and address this comment. Thank you.

Reviewer 3 Report

Comments and Suggestions for Authors

In this review, Griffith BD and Frankel TL summarized the current knowledge of AhR in cancer. The topic of the manuscript is interesting and well organized. The manuscript is generally well written.

Specific comments:

    1)    Figure 1: figure legend is too concise and should explains in a better way the content of the figure. Are the AhR-induced genes known? Including detailed insights into how AhR influences specific signaling pathways and molecular events would enhance the scientific depth of the discussion.

         2) Figure 2 and mechanism: This figure could benefit from a more in-depth exploration of the contrasting role of AhR in TME.

Author Response

Figure 1: figure legend is too concise and should explains in a better way the content of the figure. Are the AhR-induced genes known? Including detailed insights into how AhR influences specific signaling pathways and molecular events would enhance the scientific depth of the discussion.

The figure legend was updated to better reflect the content of the figure and text. As other review articles have thoroughly detailed AhR signaling pathways and induced genes, an in depth discussion was not added to the body of the text, but rather briefly described and an appropriate review article referenced.

Figure 2 and mechanism: This figure could benefit from a more in-depth exploration of the contrasting role of AhR in TME.

Additional information was added to figure 2 and to the legend to further explore the contrasting role of AhR in the TME. Thank you.

Reviewer 4 Report

Comments and Suggestions for Authors

This is a very well written review on the Aryl Hydrocarbon Receptor (AhR) and its functions in carcinogenesis and regulation of immune response. also, the authors consider and discuss the targeting of this receptor.

The topic is of interest, and it has been well presented and discussed.

The only two points I would consider are

a- the insertion in the figure 1 of the AhR ligands. The indication already present is too general, and it does not reflect what has been described in the text. Indeed, both extracellular and intracellular ligands are involved in the activation of the AhR function.

b-I would insert a figure regarding the different types of targeting of the AhR to help the reader.

Comments on the Quality of English Language

The English language is good.

Author Response

The insertion in the figure 1 of the AhR ligands. The indication already present is too general, and it does not reflect what has been described in the text. Indeed, both extracellular and intracellular ligands are involved in the activation of the AhR function.

The insertion of figure 1 into the body of the text was updated to better reflect what is described in the text, and intracellular ligands were included.

I would insert a figure regarding the different types of targeting of the AhR to help the reader.

Developing a concise additional figure to demonstrate the different types of targeting of AhR proved difficult. Clarifying statements were added to the therapeutic potential and future directions section in the hopes to address this.

Round 2

Reviewer 1 Report

Comments and Suggestions for Authors

No further comments